# Correlation of Nabiximols Dose to Steady-State Concentrations of Cannabinoids in Urine Samples from Patients with Multiple Sclerosis

**DOI:** 10.3390/jcm11133717

**Published:** 2022-06-27

**Authors:** Rüdiger Birke, Stefanie Meister, Alexander Winkelmann, Burkhard Hinz, Udo I. Walther

**Affiliations:** 1Institute of Pharmacology and Toxicology, Rostock University Medical Center, 18057 Rostock, Germany; r.birke91@web.de (R.B.); burkhard.hinz@med.uni-rostock.de (B.H.); 2Department of Neurology, Rostock University Medical Center, 18147 Rostock, Germany; stefanie.meister@med.uni-rostock.de (S.M.); alexander.winkelmann@med.uni-rostock.de (A.W.)

**Keywords:** nabiximols, patients, multiple sclerosis, drug monitoring, THC-COOH metabolite, urine assay, DRI^TM^ Cannabinoid (THC) Assay, immunological test

## Abstract

Therapeutic drug monitoring of Δ^9^-tetrahydrocannabinol (THC) and cannabidiol (CBD) is based on a complex procedure and is therefore not possible in most laboratories, especially in emergency cases. This work addresses the question of whether therapeutic drug monitoring of nabiximols can be performed using an immunological urine-based test system for cannabinoid abuse. Seventeen patients with multiple sclerosis were included in this study. Administered doses of nabiximols were correlated with immunologically determined urine concentrations of cannabinoids using the DRI^TM^ Cannabinoid (THC) Assay. Significant correlations with the administered nabiximols doses were found for creatinine-normalized urine concentrations of cannabinoids without (r = 0.675; *p* = 0.0015) and after (r = 0.650; *p* = 0.0044) hydrolysis, as well as for gas-chromatography-coupled mass spectrometry (GC/MS)-measured concentrations of the THC metabolite 11-nor-9-carboxy-Δ^9^-THC (THC-COOH) in urine samples (r = 0.571; *p* = 0.0084) by Pearson’s correlation. In addition, doses were significantly correlated with plasma THC-COOH concentrations (r = 0.667; *p* = 0.0017) measured by GC/MS. Simple immunological cannabinoid measurements in urine samples could provide an estimate of nabiximols dosage, although the correlations obtained here were weak because of the small number of patients observed. Longitudinal monitoring of individual patients is expected to exhibit good results of therapeutic drug monitoring of nabiximols.

## 1. Introduction

Nabiximols is a plant extract mixture from the leaves and flowers of the hemp plant (*Cannabis sativa* L.) containing roughly equal amounts of Δ^9^-tetrahydrocannabinol (THC) and cannabidiol (CBD). In the form of an oromucosal spray (Sativex™), nabiximols is approved in Germany for symptom improvement in adult patients with moderate to severe spasticity due to multiple sclerosis (MS) who have not responded adequately to other antispasticity drug therapy and who demonstrate clinically significant improvement in spasticity-related symptoms during an initial therapy trial. The quantification of spasticity in most previous studies addressing the efficacy of nabiximols was performed via visual analogue scaling. Therefore, in most cases, no objective measurement was used. In some reviews, good efficacy was questioned [1], while in others, a fair [2] or small [3] effect was described. In a recent study with MS patients receiving oromucosal nabiximols, a significant inverse correlation was found between the median intrasubject repeated numerical rating scale for spasticity and the corresponding median THC and CBD plasma concentrations [4].

It is commonly accepted that a cannabinoid therapy is well tolerated, even after high doses have been administered [1,5]. On the other hand, seizures have even been described with the clinical use of THC [6] or nabiximols [7], although a causal relationship for these adverse effects has not been clearly established. In addition, other adverse drug effects may occur, although abuse potential for nabiximols seems to exist only at higher doses [8]. When evaluating possible adverse drug effects, both disease-derived symptoms and drug-withdrawal-related effects must be considered as differential diagnoses. We became interested in this topic when a 58-year-old MS patient (3 years of nabiximols therapy) presented postictally to our emergency department with the question of whether or not the seizure could be nabiximols-related. Therefore, monitoring of nabiximols administration could be useful for routine use and also in some emergencies.

As some metabolites of THC have prolonged elimination half-lives, these metabolites might be useful in monitoring therapeutic application. However, CBD, THC, and the primary hydroxylated metabolites have half-lives of from a few minutes after inhalative application up to a few hours after oromucosal uptake [9,10,11,12,13]; thus, they cannot be used to monitor an appropriate dosing regimen. In many laboratories, THC abuse is monitored by simple 11-nor-9-carboxy-Δ^9^-THC (THC-COOH) analysis of urine samples using commonly available immunological methods, although more elaborate gas-chromatography-coupled mass spectrometry (GC/MS) methods for determining plasma concentrations of THC-COOH have also been described. Unfortunately, the latter procedures are not well suited for emergency situations, as GC/MS methods are not widely available, and a derivatization step is mostly required here.

This work therefore addresses the question of whether nabiximols administration can be monitored using commonly available immunology-based THC abuse testing kits for urine samples.

## 2. Materials and Methods

### 2.1. Patients

All patients were admitted to the Department of Neurology, Rostock University Medical Center, Germany, with a diagnosis of MS. All patients were co-medicated with nabiximols for at least two months at the current dosage. Eight of the seventeen patients were male, and the mean age of all patients was 53.5 ± 11.6 years (mean ± standard deviation). The initial diagnosis of MS occurred approximately 14 ± 8.7 (mean ± standard deviation) years prior. Eleven of the patients with MS were classified as secondary progressive, four were classified as primary progressive, and two were classified as relapsing–remitting. Nine patients received pulsed high-dose glucocorticoid therapy, five received mitoxantrone, and one each received betaferon, fingolimod, alemtuzumab, or azathioprine in addition to nabiximols. Eight patients received additional fampridine, and two received tizanidine. Gabapentin or pregabalin was administered in five patients, and amantadine was administered in two patients. Standard antidepressive therapy was given to five patients, pain medication to two, dopaminergic therapy to three, and a standard antihypertensive combination to seven.

### 2.2. Sample Preparation

Blood and urine samples were collected at least 2 h after the last nabiximols application, as indicated by the patients. Urine samples were hydrolyzed enzymatically with β-glucuronidase (type HP-2, 2500 U/mL for 12 h at 37 °C, pH 4.5) as described by Bergamaschi et al. [14].

### 2.3. Immunological Determination

Cannabinoids were quantified in both urine samples (after hydrolysis and without) by an immunological method. The immunological method was performed on a Cobas Mira Plus system using the DRI^TM^ Cannabinoid (THC) Assay purchased from Microgenics Corp. (Fremont, CA, USA). The assay was performed as recommended by the manufacturer. The assay is not specific for THC-COOH. THC, 11-nor-Δ^8^-THC-COOH, 11-hydroxy-Δ^9^-THC (11-OH-THC), 8β-hydroxy-Δ^9^-THC (8β-OH-THC), 8β,11-dihydroxy-Δ^9^-THC (8β,11-diOH-THC), and, to a much lesser extent, cannabinol were found to exhibit cross-reactivity with the assay (manufacturer´s description). Creatinine concentrations were measured using the Cobas Mira Plus system and the DRI^TM^ Creatinine-Detect Test from Microgenics Corp. (Fremont, CA, USA).

### 2.4. GC/MS Analysis

CBD, THC, 11-OH-THC, and THC-COOH were measured by GC/MS in all plasma, urine, and hydrolyzed urine samples. In each case, determination by GC/MS was performed after triple liquid/liquid extraction (hexane/ethyl acetate ratio 7/1) in alkaline conditions (pH 10) for CBD, THC, and 11-OH-THC or twice under acidic conditions (pH 4) for THC-COOH. The extract was dried under nitrogen flow, and the residue was silylated with N-methyl-N-(trimethylsilyl)-trifluoroacetamide (MSTFA) (60 °C, 30 min). Chromatographic determination was performed on a 5890 GC Series II combined with a 5971 mass selective detector (MSD), both from Hewlett Packard (Palo Alto, CA, USA), at 280 °C injection temperature and 280 °C MSD interface temperature using an HP-1MS column (12 m; 0.2 mm inner diameter, 0.33 µm film thickness). The oven temperature started at 70 °C (constant for 3 min), then increased 30 °C per min up to 300 °C and remained constant at 300 °C for another 5 min.

D3-CBD, D3-THC, D3-11-OH-THC, and D3-THC-COOH (all from Cerilliant, Round Rock, TX, USA) were used as internal standards. External standard calibration curves were prepared for each substance (also obtained from Cerilliant) between 1 and 100 ng/mL (except for THC-COOH in urine, between 5 and 500 ng/mL) each as calibration curves by linear regression. The mean correlation coefficients of all calibration curves ranged from 0.999 to 0.978 for plasma curves (*n* = 10) and from 0.999 to 0.966 for urine samples (*n* = 10) for CBD, THC, 11-OH-THC, and THC-COOH. The spiked and corresponding calculated concentrations of the calibration samples are shown in Figure 1 using the example of THC in plasma samples. The lower limit of quantification (LOQ) was accepted if its deviation from the mean value of the corresponding calibrator from different runs was less than 20%. LOQs determined for the GC/MS method were at 1 ng/mL (11-OH-THC in urine, THC in plasma), 3 ng/mL (CBD in urine and plasma, THC in urine, 11-OH-THC and THC-COOH in plasma), and 5 ng/mL (THC-COOH in urine). Urine from healthy male volunteers, all free of drugs of abuse, was used for these calibration curves. Plasma was provided by the Institute of Transfusion Medicine at Rostock University Medical Center and was also free of drugs (except caffeine).

### 2.5. Statistics

Statistical analysis of nabiximols doses and substance concentrations was performed using Pearson product-moment correlation, with *p* ≤ 0.05 as the significance threshold. Normal distribution of the data was confirmed by Shapiro–Wilk test.

## 3. Results

Cannabinoids were measured in urine samples using an immunological assay to establish a simple method for monitoring nabiximols therapy. A significant relationship was found between the immunologically determined (and normalized to creatinine) cannabinoid concentrations of the urine samples and the daily nabiximols doses by Pearson correlation coefficients (Figure 2A: without hydrolysis: r = 0.675; Figure 2B: after hydrolysis: r = 0.650). A similar result was obtained after measuring THC-COOH concentrations by GC/MS, with a significant correlation to daily doses of nabiximols for the non-hydrolyzed urine samples (Figure 2C: r = 0.571) and the hydrolyzed samples (r = 0.621; not shown), as well as for the plasma samples (Figure 2D: r = 0.667).

To confirm the required time interval of at least 2 h between the last nabiximols administration and the collection of the blood and urine samples, CBD, THC, and 11-OH-THC were measured by GC/MS in addition to THC-COOH in all samples. According to Table 1, the plasma levels of THC and the carboxylated metabolite of THC were mostly detectable well above their LOQ (THC: 1 ng/mL, THC-COOH: 3 ng/mL), whereas the concentrations of CBD and most of the hydroxylated metabolites of THC were below their LOQ (of 3 ng/mL). Strikingly, almost no CBD or THC was found in the non-hydrolyzed urine samples (concentrations below the LOQ of 3 ng/mL and even below the detection limit of 1 ng/mL). Likewise, 11-OH-THC was mostly below the LOQ of 1 ng/mL. Only THC-COOH could be measured above the LOQ of 5 ng/mL in most urine samples.

Referring to the literature, the ratio of THC-COOH/THC can be used to assess the period between THC application and the collection of a plasma sample [15,16]. Accordingly, Table 2 shows the ratios of the THC-COOH and THC concentrations in the plasma samples presented in Table 1. Most of the ratios are well above a value of 10, with only three ratios below a value of 2 to 5.

In addition, the ratios of hydrolyzed and non-hydrolyzed metabolites in urine samples are often used to assess the last ingestion of THC [17]. In fact, even higher concentrations of cannabinoid substances were found after hydrolysis by GC/MS. Accordingly, values up to about 200 µg/L were measured for CBD, but in one-third of all samples, CBD was not elevated above the LOQ (not shown). For THC, levels above the LOQ were detected in only one-third of all samples (not shown). The ratios of the concentrations of THC-COOH and 11-OH-THC in hydrolyzed and non-hydrolyzed urine samples are shown in Table 3, with maximum increases of 20.9 (THC-COOH) and >34.4 (11-OH-THC). In contrast, the measurement of cannabinoid concentrations using the immunological assay system showed decreased concentrations after the hydrolysis procedure (Table 3).

**Table 2 jcm-11-03717-t002:** THC-COOH/THC concentration ratios in plasma samples determined by GC/MS. Ratios were calculated by forming the quotients of the THC-COOH and THC concentrations of each plasma sample. NA: not applicable, as both concentrations were below the LOQ.

Patient Number	THC-COOH/THC Ratio
1	>3.6
2	>11.5
3	NA
4	62.22
5	250.5
6	>22.3
7	12.1
8	>10.6
9	<0.67
10	44.2
11	12.7
12	20.2
13	9.3
14	14.5
15	1.07
16	0.76
17	>16.3

## 4. Discussion

The medical use of cannabinoids (THC and CBD) is mostly described as a well-tolerated therapeutic option [18,19,20,21,22,23,24,25]. Nevertheless, some adverse drug effects have been described in connection with the use of THC and nabiximols, including seizures [6,7], although for this particular case, no causal relationship to the cannabinoids administered has been proven so far. Especially in case of such serious effects, the appropriate use of the cannabinoid preparation should be reviewed with high priority. Therefore, a common method for therapeutic drug monitoring might be useful. In this work, we addressed the question of whether the administration of nabiximols can be monitored by immunological methods on urine samples for THC, because these methods are very common in clinical toxicology, so such therapeutic drug monitoring could be widely performed.

Because THC-COOH has an elimination half-life of approximately 3–11 days with frequent abuse of THC [26], a very consistent urinary elimination is expected, so the measurement of urine concentrations may also be useful in assessing steady-state plasma concentrations. Therefore, a THC-COOH-selective method is expected to be valuable in assessing the steady-state cannabinoid concentrations under nabiximols therapy. As our data show, the selective measurement of THC-COOH by GC/MS correlated with patient-reported nabiximols doses. Moreover, the correlation of the values obtained by the immunological method was in the same range, although the method was not selective for THC-COOH. THC, 11-nor-Δ^8^-THC-COOH, 11-OH-THC, 8β-OH-THC, and 8β,11-diOH-THC were also found to react positively in the immunoassay. Moreover, a significantly lower concentration of cannabinoids was measured by the immunological method after hydrolysis compared to the condition before. This suggests that besides the described metabolites, hydrolysable metabolites were already measured with this immunological system under the conditions of non-hydrolysis, which could explain the divergent findings obtained with both methods. Therefore, in patients with different metabolism patterns, weak correlations of doses and measured concentrations should be expected. On the other hand, the correlation of different doses for an individual patient should be meaningful.

Patients´ statements about administered doses of their medications are always questionable. Some patients want to avoid the side effects of a drug, or, for example, in the case of nabiximols, want to avoid conflicts with driving ability. Others expect better effectiveness when a recommended dose is increased. Therefore, monitoring of drug levels may be useful. This is often mandatory if the physician is considering a change in drug dosing regimen. Significant correlations for administered doses and plasma levels are usual under steady state conditions. Because THC and CBD are eliminated with short half-lives [10,11], steady-state conditions cannot be achieved for these substances; the same is true for 11-OH-THC, for the same reason. Concentrations of these substances in urine samples are also short-lived; higher concentrations can be expected only with a recently applied dose. In our study, a period of at least 2 h was demanded between the last application of nabiximols and the collection of blood and urine samples from the patients. Accordingly, no measurable and, in particular, no higher concentrations of THC, CBD, or 11-OH-THC were expected in the plasma samples, whereby a higher concentration might indicate more frequent use by the patients than reported.

A well-known criterion for assessing the interval since last smoking of marijuana is the THC-COOH/THC ratio [15,16]. Because oromucosal applications might have similar pharmacokinetic properties, this THC-COOH/THC ratio might also indicate the time interval since the last nabiximols application. This would require a reliable measurement of low THC concentrations in a range below 1 ng/mL (as expected approximately >2 h after application of nabiximols). Such measurements may be possible especially in the scientific field, but this does not seem to be practicable for routine laboratory use due to the complex analytical procedure. Considering this THC-COOH/THC ratio for an interval of at least 2 h between application and sample collection, a ratio greater than approximately 2–5 should be expected. Assuming no altered resorption kinetics in MS patients (e.g., due to a damaged oral mucosa in later stages of MS), an incorrect time interval might be assumed only in three (patient numbers 9, 15, 16) of our patients (Table 2).

The ratio of cannabinoids including hydrolysis products is also used to allow an assessment of the time period between cannabinoid administration and sample collection [17]. As in the study by Abraham et al. [17], a high efficiency of enzymatic hydrolysis was found for 11-OH-THC and a poor one for THC-COOH in our samples. Moreover, in agreement with Bergamaschi et al. [14], a high hydrolysis efficiency of CBD-derived metabolites was confirmed. Unfortunately, many of the concentrations measured in our samples were below the LOQ, so no reliable ratios could be calculated at this point.

THC-COOH has a longer half-life than the other tested substances and could therefore be a suitable candidate for monitoring steady-state cannabinoid levels under nabiximols therapy. In the work of Karschner et al. [10], two differently dosed applications of nabiximols were compared with respect to concentration–time courses for CBD, THC, and 11-OH-THC. Low concentrations were measured in plasma samples, with a concentration maximum approximately 3 h after application. Concentrations decreased with half-lives of about 3 h for all three compounds. In their work, as in ours, higher concentrations of THC were found compared with CBD, whereas 11-OH-THC was in the range of THC concentrations. Very similar results were published by Stott et al. [11], although in their work, the maximum concentrations were reached at about 1 h after nabiximols application (in their control groups). Such a very short time to maximal concentrations for THC was also described by Indorato et al. [13], where, in their study, the mean concentration of THC after application of one nabiximols puff was approximately 0.5 ng/mL.

Finally, we have no reason to believe that our patients’ statements about their dosing regimens are incorrect. Therefore, the rather weak correlation of the nabiximols doses with the measured cannabinoid concentrations could be due to differences in the metabolism of the patients, possibly resulting from different concomitant medications. In conclusion, simple immunological cannabinoid measurements in urine samples could provide an estimate of the nabiximols dose. It is expected that meaningful results could be obtained via this approach, especially in longitudinal follow-up of individual patients.

## Figures and Tables

**Figure 1 jcm-11-03717-f001:**
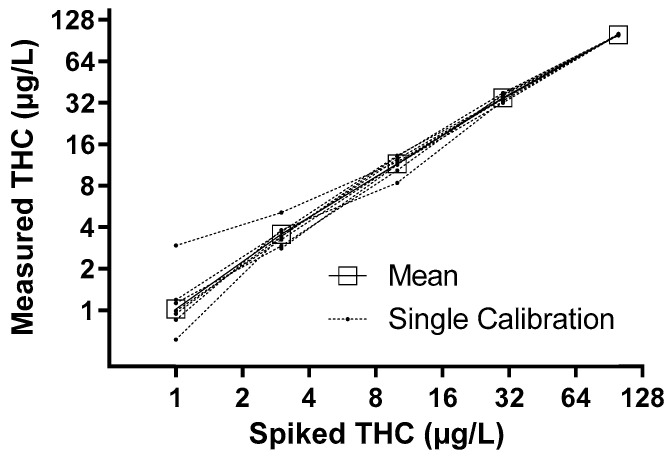
Spiked and calculated THC concentrations after linear regression of measured values of different runs. The THC values were determined by GC-MS in plasma.

**Figure 2 jcm-11-03717-f002:**
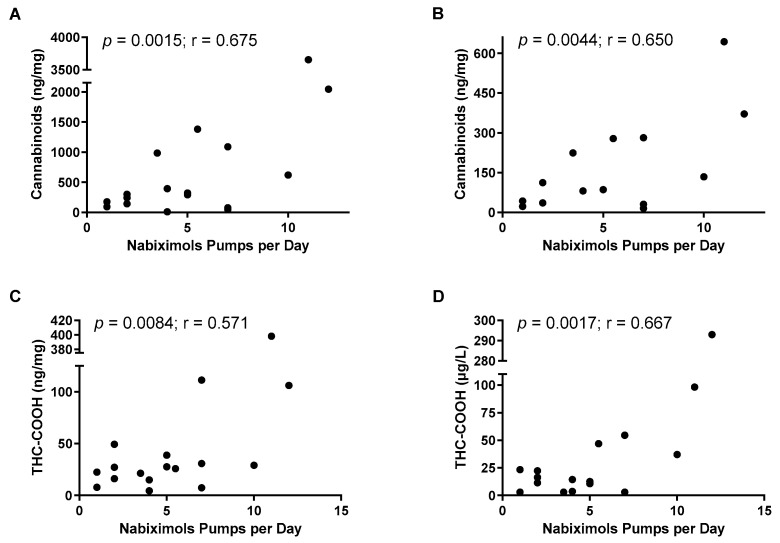
Correlations of cannabinoid concentrations versus nabiximols doses in MS patients. Cannabinoid concentrations were assessed by the immunological method in urine samples after normalizing to creatinine in non-hydrolyzed (**A**) or hydrolyzed (**B**) samples. Specific THC-COOH was measured by GC/MS in non-hydrolyzed urine samples after normalizing to creatinine (**C**) or in plasma (**D**). Statistical analysis was performed using Pearson’s product-moment correlation.

**Table 1 jcm-11-03717-t001:** Cannabinoid concentrations measured by GC/MS in plasma and urine samples of MS patients. CBD, THC, 11-OH-THC, and THC-COOH were measured by GC/MS in plasma and in non-hydrolyzed urine samples. Plasma or urine samples (1.0 mL) were extracted as described, and CBD, THC, 11-OH-THC, and THC-COOH were determined by GC/MS analysis after silylation with N-methyl-N-(trimethylsilyl)-trifluoroacetamide.

Patient Number	Cannabinoid Plasma Concentration (µg/L)				Cannabinoid Urine Concentration (µg/L)			
	CBD	THC	11-OH-THC	THC-COOH	CBD	THC	11-OH-THC	THC-COOH
1	<3	<1	<3	3.59	<3	<3	<1	<5
2	<3	<1	<3	11.5	<3	<3	1.11	37.5
3	<3	<1	<3	<3	7.08	<3	1.25	37.6
4	<3	1.17	<3	72.8	3.54	<3	<1	<5
5	<3	4.61	3.7	1155	<3	<3	<1	30
6	<3	<1	<3	22.3	<3	<3	<1	17.9
7	<3	24.2	7.73	293	<3	<3	<1	144
8	<3	<1	<3	10.6	<3	<3	<1	<5
9	<3	4.46	<3	<3	3.83	<3	<1	8.72
10	<3	1.06	<3	46.9	<3	<3	<1	32.1
11	<3	1.84	<3	23.3	<3	<3	1.7	<5
12	4.89	4.85	14.2	98.2	<3	<3	1.29	173
13	<3	3.98	<3	37.1	<3	<3	<1	48.3
14	<3	3.77	3.75	54.5	5.52	4.58	3.49	116
15	<3	11.8	<3	12.6	<3	<3	<1	11.3
16	<3	18.9	9.82	14.3	<3	<3	<1	6.53
17	<3	<1	<3	16.3	<3	<3	<1	<5

**Table 3 jcm-11-03717-t003:** Ratios of cannabinoid concentrations after hydrolytic and non-hydrolytic processing of urine samples. Columns 2 and 3 show the ratios of concentrations measured by GC/MS, while Column 4 shows the ratio of cannabinoid concentrations measured by the immunological method. NP: hydrolysis not performed; NQ: not quantifiable (value after hydrolysis below LOQ).

Patient Number	THC-COOH Ratio (Hydrolyzed/Non-Hydrolyzed); GC/MS	11-OH-THC Ratio (Hydrolyzed/Non-Hydrolyzed); GC-MS	Cannabinoid Ratio (Hydrolyzed/Non-Hydrolyzed); Immunoassay
1	NP	NP	NP
2	NQ	19.9	NP
3	NQ	12.4	0.805
4	>2.44	NQ	0.971
5	10.8	>6.9	0.455
6	2.7	NQ	0.746
7	5.7	>32	0.363
8	>3.9	NQ	0.343
9	2.9	NQ	0.359
10	20.9	>32.6	0.403
11	>4.7	2.9	0.479
12	6.1	77	0.347
13	10.8	>34.4	0.433
14	3.8	16.5	0.516
15	2.6	NQ	0.587
16	8	>7.8	0.409
17	>3.5	NQ	0.459

## Data Availability

The data presented in this study are available on reasonable request from the corresponding author.

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
