# Peer review of "Correlation of Nabiximols Dose to Steady-State Concentrations of Cannabinoids in Urine Samples from Patients with Multiple Sclerosis"

_jcm, 2022, doi:10.3390/jcm11133717_

Round 1
Reviewer 1 Report
Statistical section:
Authors used Pearson's correalation, method which is used for data with normal distribution.
Did the authors check the distribution of the data and with which test?
P values should be presented in the abstract, as least at p<0.001
Reviewer 2 Report
The study evaluate whether CBD and THC can be reliably detect in urine with a simple immunological urine-based test in patients with MS treated with nabiximols. Addressing this point is of particular interest, as the drug efficacy is usually assessed with a patient self reported system (NRS). To strenghten this concept, I suggest to the authors to underline in the introduction/discussion the fact that some recent studies have shown a good correlation between nabiximols plasma metabolites levels and clinical effecctiveness on spasticity in MS. (see Contin et al. Clinical Neuropharmacology, 2018) Furthermore, preliminary evidence suggest a measurable effect on gait and motor performances. Lastly, I have 2 other minor points:
- can the authors specify the median time post dosing of blood and urine samples?
- did the authors evaluated whether there was a relationship between CBD/THC urine levels and the degree of spasticity improvement (e.g. NRS decrease) after nabiximols dosing?
Reviewer 3 Report
The study addresses the large unmet need of quickly assessing cannabinoid levels in Nabiximols-treated patients. This could be of potential therapeutic and safety use. Nevertheless conclusions are limited by the results that fall short of proving their initial aim. The finding that higher urinary cannabinoid levels at an immunological test (badly) correlate with the prescribed dose of Nabiximols, is per se insufficient to make a case that it can be used in clinical practice to estimate Nabiximol dosage. I.e. non-hydrolyzed cannabinoid <500 ng/mL correspond to Nabiximols doses 1-7/day, and hydrolyzed dose <150 ng/mL correspond to Nabiximols doses 1-10. What is the clinical utility of this measurement? There is no formal validation of the immunological methods towards the GC/MS (no ICC?).
As a minor point, epilepsy has not been clearly linked to cannabinoid or Nabiximols use. Few reports do not allow to consider it as a side effect. In addition to this, cannabis extracts containing THC, have been successfully used to treat drug-resistant epilepsy.
I would propose revising the manuscript, tampering down conclusions based on the proposed results, compare immunologically determined measurements to GC/MS, and resubmit.
Round 2
Reviewer 1 Report
Dear atuhros,
thank you for your corrections.
Manuscript is suitalbe for publication in present form.